# Adherence to practice parameters in Medicare beneficiaries with amyotrophic lateral sclerosis

Osvaldo J. Laurido-Soto[1]*, Irene M. Faust[1,2], Susan Searles Nielsen[1], Brad A. Racette[1,2,3]

1 Department of Neurology, Washington University School of Medicine, Saint Louis, Missouri, United States of America, 2 Department of Neurology, Barrow Neurological Institute, Phoenix, Arizona, United States of America, 3 School of Public Health, Faculty of Health Sciences, University of the Witwatersrand, Johannesburg, South Africa

* ojlaurido-soto@wustl.edu

## Abstract

### Objective

Physician adherence to evidence-based clinical practice parameters impacts outcomes of amyotrophic lateral sclerosis (ALS) patients. We sought to investigate compliance with the 2009 practice parameters for treatment of ALS patients in the United States, and sociodemographic and provider characteristics associated with adherence.

### Methods

In this population-based, retrospective cohort study of incident ALS patients in 2009–2014, we included all Medicare beneficiaries age ≥20 with ≥1 International Classification of Diseases, Ninth Revision, Clinical Modification ALS code (335.20) in 2009 and no prior years (N = 8,575). Variables of interest included race/ethnicity, sex, age, urban residence, Area Deprivation Index (ADI), and provider specialty (neurologist vs. non-neurologist). Outcomes were use of practice parameters, which included feeding tubes, non-invasive ventilation (NIV), riluzole, and receiving care from a neurologist.

### Results

Overall, 42.9% of patients with ALS received neurologist care. Black beneficiaries (odds ratio [OR] 0.56, 95% confidence interval [CI] 0.47–0.67), older beneficiaries (OR 0.964, 95% CI 0.961–0.968 per year), and those living in disadvantaged areas (OR 0.70, 95% CI 0.61–0.80) received less care from neurologists. Overall, only 26.7% of beneficiaries received a feeding tube, 19.2% NIV, and 15.3% riluzole. Neurologist-treated patients were more likely to receive interventions than other ALS patients: feeding tube (OR 2.80, 95% CI 2.52–3.11); NIV (OR 10.8, 95% CI 9.28–12.6); and riluzole (OR 7.67, 95% CI 6.13–9.58), after adjusting for sociodemographics. These associations remained marked and significant when we excluded ALS patients who subsequently received a code for other diseases that mimic ALS.

**Data Availability Statement:** Due to contracting requirements with the Centers for Medicare and Medicaid Services (CMS), the data and associated

materials cannot be made publicly available. This type of data, which is de-identified research data with selected information at the individual level is obtained from CMS through a formal Data Use Agreement with CMS via a secure, CMS-approved data access method. The detailed data we used for the present dataset is a custom dataset, as detailed in the Methods. All summary data that we are able to share is contained in the paper. Information and inquiries on how to obtain CMS data, with specific details about the process for requesting data and contact information for initiating the process through the Research Data Assistance Center (ResDAC) can be found at https://resdac.org.

**Funding:** This study was funded from the Hope Center for Neurological Disorders (BAR, SSN), NIH – NIEHS (K01ES028295 (SSN)), Tambourine/ Northeast ALS Consortium (BAR, SSN), Paula & Rodger Riney Charitable Fund (BAR, SSN), the Kemper and Ethel Marley Foundation (BAR), and Washington University School of Medicine Faculty Diversity Scholars Program (OJLS). The funders had no role in study design, data collection and analysis, decision to publish, or preparation of the manuscript.

**Competing interests:** The authors have declared that no competing interests exist.

## Conclusions

ALS patients treated by neurologists received care consistent with practice parameters more often than those not treated by a neurologist. Black, older, and disadvantaged beneficiaries received less care consistent with the practice parameters.

## Introduction

Life expectancy in amyotrophic lateral sclerosis (ALS) is short, with average survival two years after diagnosis [1], and only 10% survival after five years [2]. Interventions that improve survival and quality of life are critical for the care of these patients. In October 2009, the American Academy of Neurology (AAN) published practice parameters for ALS patient care with evidence-based interventions that improve ALS patients' survival and quality of life. These recommended interventions included prescription of riluzole, early feeding tube placement, and non-invasive ventilation (NIV) [3]. European guidelines also support these practice parameters [4]. Implementation of guidelines can be slow and dependent on provider education [5, 6]; although, these evidence-based practice parameters and care by neurologists in specialised ALS centers improve outcomes [7, 8], potentially due to guideline adherence.

Unfortunately, access to neurologist care is constrained by demographic, geographic, and social determinants of health (SDOH) barriers to care. Black, Hispanic, and Native American patients are less likely to see an outpatient neurologist for most common neurological diseases [9], and access less frequently high-volume stroke centers, compared to non-minoritized populations [10, 11]. Black patients also have longer diagnostic delays [12], suggesting barriers to tertiary care could potentially delay evidence-based disease interventions. We sought to identify adherence to practice parameters for ALS patients in the United States (U.S.) and determine if there were differences based on provider specialty and patient sociodemographics.

## Methods

### Standard protocol approvals

The study was approved by the Washington University in St. Louis Human Research Protection Office and the Centers for Medicare and Medicaid Services (CMS). All data were de-identified prior to release by CMS.

### Study population and data sources

We conducted a population-based, retrospective cohort study of Medicare beneficiaries age ≥20 diagnosed with ALS in 2009. We obtained comprehensive Medicare claims data from 2004–2014 to ensure incident diagnoses and followed these incident ALS patients forward in time. Medicare is the only national health care system available to all age-eligible citizens or permanent legal residents in the U.S. and used by most adults age 65 and older. In addition, Medicare coverage is available regardless of age once a patient receives a diagnosis of ALS. For this study, we identified and included all incident ALS cases in 2009 who met the following study eligibility criteria, as determined using the Beneficiary Annual Summary File (BASF) from 2009: 1) age ≥20 years, 2) residence in the 50 U.S. states or District of Columbia, and 3) enrolled in Medicare Parts A and/or B without Part C (e.g. Health Maintenance Organization (HMO)), coverage. Optional Part D (pharmacy) coverage was only a requirement for the analyses investigating use of riluzole. The only study in the U.S. to assess the feasibility of using

administrative data to identify ALS cases observed that HMO coverage limited ascertainment [13], so we applied the Medicare coverage-based restrictions noted above to facilitate complete ascertainment of cases, as well as their utilization of interventions. We used comprehensive claims data and BASFs for 2010–2014 to follow cases up to six years following diagnosis to assess intervention utilization, neurologist care, and date of death. Informed consent was not required in this records-based study, which was classified as not involving human subjects research by the Washington University in St. Louis Human Research Protection Office; we obtained a waiver.

## Identification of ALS

CMS identified beneficiaries with ≥1 International Classification of Diseases, Ninth Revision, Clinical Modification (ICD-9-CM) motor neuron disease (MND) code (335.2x) in 2004–2009 in inpatient, outpatient, Part B physician/carrier, skilled nursing facility, home health care, durable medical equipment, and hospice files. To date, only one population-based study assessed accuracy of administrative databases for identifying ALS and reported 93.79% sensitivity and 99.97% specificity, when requiring ≥1 diagnosis code for MND in hospital discharge or health insurance data [14]. Two hospital-based studies in Italy found nearly identical results [15, 16]. Accordingly, we used detailed claims data to identify those with ≥1 ICD-9-CM ALS code (335.20) in 2009 but no prior MND code, (335.2x), i.e., incident ALS cases. A study with administrative data from the U.S. found that this criterion of ≥1 ICD-9-CM 335.20 code was 96% sensitive and 52% specific for differentiating ALS from other MND, with a substantially more complex algorithm only shifting the balance between sensitivity (85%) and specificity (87%) [13]. Using the simpler, high sensitivity approach to ensure that true ALS (vs. other MND) cases were representative demographically, we identified 8,583 incident cases. We excluded eight with missing demographic information, resulting in 8,575 incident ALS cases for analysis. As an alternative case definition for sensitivity analyses, we excluded those patients that had an ICD-9-CM code consistent with a potential ALS/MND mimic [17] after they had received a 335.2x code to minimize the possibility of inappropriately miss-accounting differences in practice parameter utilization in those patients whose diagnosis changed after the initial ALS diagnosis. These potential mimics included structural spinal pathologies (spondylosis and allied disorder– 721.0-4x, spinal stenosis in cervical region– 723.0, and spinal stenosis of other regions– 724.0x), myasthenia gravis and subtypes (358.0x, 358.1, 358.3x, 358.8, 358.9), polyneuropathies (polyneuropathy in other diseases classified elsewhere, including multifocal motor neuropathy– 357.4, diabetic polyneuropathy– 357.2, hereditary spastic neuropathy– 356.x, chronic inflammatory demyelinating polyneuropathy– 357.81), and spinal muscular atrophies (335.1x). We were unable to assess for spinobulbar muscular atrophy/Kennedy's disease (Current Procedural Terminology [CPT] codes– 81173, 81174, 81204), as these codes were not accepted until 2019. We did not exclude patients with a code for another MND as these diseases have a common phenotype and may be difficult to fully differentiate [18]. In addition, based on prior administrative data algorithms [13], and the reported low misclassification by neurologist [17, 19, 20], we retained as true ALS cases those patients who received an ALS code from a neurologist. Utilizing this alternative case definition, we identified 6,888 (80.3%) of incident ALS cases for sensitivity analysis, representing all 3,676 ALS cases who received care from a neurologist and 3,212 of 4,899 ALS cases who did not.

## Healthcare utilization variables

We used comprehensive Medicare medical procedure claims from 2009–2014 to identify ever use of each of the three ALS interventions specified in the practice parameters.

1. NIV: ≥1 Healthcare Common Procedure Coding System (HCPCS) code for devices with bi-level capabilities (E0464, E0470, E0471) [21, 22]. We excluded ICD-9-CM procedure 93.9x for non-invasive ventilation, as it is predominantly an inpatient code and may not reflect true outpatient utilization.

2. Feeding tube: ≥1 ICD-9-CM procedure codes (43.1, 43.11, 43.19, 44.32) or CPT codes (43246, 43750, 43653, 43830, 43832, 44372, 44373, 49440, 49441, 74350) [23–25]. We included inpatient codes for feeding tubes as we anticipate the code reflects placement of a permanent feeding tube that would only occur once.

3. Riluzole: ≥1 Medicare Part D fill of riluzole from 2010–2014.

### Ascertainment of demographic variables

Demographic variables of interest included, race/ethnicity, sex, age, urban residence, Area Deprivation Index (ADI) as a proxy for SDOH, and smoking status (see below). We obtained beneficiary date of birth (age), sex, race/ethnicity, and residential zip code using the BASF for 2009, which we linked to a rural-urban commuting area [26] (RUCA) and the ADI [27]. We also calculated distance to nearest multidisciplinary ALS clinic [28]. Finally, because smoking is associated with survival and potentially all three interventions, we classified beneficiaries as smokers if they had ≥1 tobacco-specific code (ICD-9-CM diagnosis V15.82, 305.1; CPT 99406, 99407) or ≥1 code for chronic obstructive pulmonary disease.

### Statistical analysis

We used Stata version 14.2 [29] for all analyses. We conducted logistic regression to obtain odds ratios (ORs) and 95% confidence intervals (CIs) to compare differences in sex, race/ethnicity, age, urban/rural residence, distance from residence to a multidisciplinary ALS clinic, and residence in a disadvantaged area (ADI score >80 out of 100) between beneficiaries who received care from a neurologist and those who did not. We defined care from a neurologist as ≥1 ALS diagnosis code from a neurologist any time between ALS diagnosis (first ALS code) and the end of follow-up, which was either death or December 31, 2014, whichever occurred sooner. We examined the effect of adjusting all ORs and CIs for all other variables of interest. We then used logistic regression to calculate OR and 95% CI with either NIV, feeding tube, or riluzole as the outcome and ever being treated by a neurologist as our independent variable, adjusting for all other variables. For the riluzole analyses, because Medicare Part D data in 2009 were not available to us, we required all cases to have Part D coverage and to survive to 2010 (N = 4,050, 47%). Finally, we examined the association between all sociodemographic variables and use of each of the practice parameters, with the latter as the outcomes, overall and while stratifying by 'ever/never' receiving neurologist care. Where pairs of the above associations suggested neurologist care as a possible mediator of the relationship between sociodemographics and use of an intervention, we conducted mediation analyses using 'medeff' [30]. We performed a sensitivity analysis with our alternative case definition to assess the effects of excluding those beneficiaries with a potential ALS mimic diagnosis.

## Results

### Patient demographics

Our cohort was composed primarily of non-Hispanic White (86.7%) beneficiaries with an average age of 68.1 years (standard deviation [SD] 12.9) of which 54.5% were male (Table 1), a

**Table 1. Incident ALS patient cohort demographics, and association with receipt of care from neurologist, U.S. Medicare 2009.**

| | All N = 8,575 n (%) | Received care from a neurologist N = 3,676 n (%) | Only received care from non-neurologists N = 4,899 n (%) | Unadjusted OR (95% CI)[a] | Mutually adjusted[b] OR (95% CI)[a] | Excluding ALS mimics N = 6,888 Mutually adjusted[c] OR (95% CI)[a] |
|---|---|---|---|---|---|---|
| **Sex** | | | | | | |
| Male | 4,670 (54.5) | 2,066 (56.2) | 2,604 (53.2) | 1.00 (Reference) | 1.00 (Reference) | 1.00 (Reference) |
| Female | 3,905 (45.5) | 1,610 (43.8) | 2,295 (46.9) | 0.88 (0.81–0.96) | 1.01 (0.92–1.10) | 1.04 (0.94–1.15) |
| **Race/ethnicity** | | | | | | |
| Non-Hispanic White | 7,432 (86.7) | 3,252 (88.5) | 4,180 (85.3) | 1.00 (Reference) | 1.00 (Reference) | 1.00 (Reference) |
| Black | 682 (8.0) | 216 (5.9) | 466 (9.5) | 0.60 (0.50–0.71) | 0.56 (0.47–0.67) | 0.60 (0.49–0.73) |
| Hispanic | 149 (1.7) | 70 (1.9) | 79 (1.6) | 1.14 (0.82–1.58) | 1.02 (0.71–1.46) | 0.96 (0.65–1.43) |
| Other[d] | 312 (3.6) | 138 (3.8) | 175 (3.6) | 1.02 (0.81–1.28) | 0.92 (0.73–1.17) | 0.89 (0.68–1.16) |
| | | | | overall p = 0.81 | overall p = 0.17 | overall p = 0.09 |
| **Age, years** | | | | | | |
| Mean (SD) | 68.1 (12.9) | 64.8 (12.2) | 70.6 (12.9) | 0.964 (0.961–0.968)[e] | - | - |
| 20–39 | 216 (2.5) | 117 (3.2) | 99 (2.0) | 0.93 (0.69–1.25) | 0.97 (0.72–1.31) | 0.89 (0.64–1.23) |
| 40–49 | 601 (7.0) | 348 (9.5) | 253 (5.2) | 1.08 (0.88–1.33) | 1.13 (0.92–1.40) | 1.18 (0.93–1.50) |
| 50–54 | 500 (5.8) | 267 (7.3) | 233 (4.8) | 0.90 (0.73–1.12) | 0.93 (0.75–1.16) | 0.86 (0.68–1.10) |
| 55–59 | 659 (7.7) | 348 (9.5) | 311 (6.4) | 0.88 (0.72–1.07) | 0.90 (0.74–1.11) | 0.93 (0.74–1.16) |
| 60–64 | 977 (11.4) | 547 (14.9) | 430 (8.8) | 1.00 (Reference) | 1.00 (Reference) | 1.00 (Reference) |
| 65–69 | 1,401 (16.3) | 650 (17.7) | 751 (15.3) | 0.68 (0.58–0.80) | 0.68 (0.57–0.80) | 0.76 (0.63–0.92) |
| 70–74 | 1,368 (16.0) | 614 (16.7) | 754 (15.4) | 0.64 (0.54–0.76) | 0.63 (0.53–0.75) | 0.72 (0.59–0.86) |
| 75–79 | 1,154 (13.5) | 425 (11.6) | 729 (14.9) | 0.46 (0.39–0.55) | 0.45 (0.38–0.54) | 0.52 (0.43–0.64) |
| 80–84 | 940 (11.0) | 250 (6.8) | 690 (14.1) | 0.29 (0.24–0.35) | 0.28 (0.23–0.34) | 0.29 (0.23–0.36) |
| ≥85 | 759 (8.9) | 110 (3.0) | 649 (13.3) | 0.13 (0.11–0.17) | 0.13 (0.10–0.17) | 0.13 (0.10–0.17) |
| | | | | $p_{trend} < 0.001$ | $p_{trend} < 0.001$ | $p_{trend} < 0.001$ |
| **Urban-rural residence[f]** | | | | | | |
| Metro/micropolitan | 7,790 (91.0) | 3,317 (90.3) | 4,473 (91.5) | 1.00 (Reference) | 1.00 (Reference) | 1.00 (Reference) |
| Small town/rural | 772 (9.0) | 356 (9.7) | 416 (8.5) | 1.15 (1.00–1.34) | 1.09 (0.93–1.28) | 1.07 (0.89–1.28) |
| **Miles to ALS clinic[g]** | | | | | | |
| 0–49 | 4,749 (55.7) | 1,962 (53.7) | 2,787 (57.2) | 1.00 (Reference) | 1.00 (Reference) | 1.00 (Reference) |
| 50–99 | 1,739 (20.4) | 776 (21.2) | 963 (19.8) | 1.15 (1.03–1.28) | 1.11 (0.99–1.25) | 1.15 (1.01–1.31) |
| 100–149 | 1,043 (12.2) | 471 (12.9) | 572 (11.7) | 1.17 (1.02–1.34) | 1.15 (0.99–1.32) | 1.22 (1.04–1.44) |
| >150 | 997 (11.7) | 446 (12.2) | 551 (11.3) | 1.15 (1.00–1.32) | 1.09 (0.94–1.26) | 1.17 (0.99–1.38) |
| | | | | $p_{trend} = 0.88$ | $p_{trend} = 0.45$ | $p_{trend} = 0.88$ |
| **ADI[h]** | | | | | | |
| ADI disadvantaged | 1,312 (15.5) | 475 (13.2) | 837 (17.3) | 0.73 (0.64–0.82) | 0.70 (0.61–0.80) | 0.73 (0.63–0.84) |
| ADI not disadvantaged | 7,141 (84.5) | 3,131 (86.8) | 4,010 (82.7) | 1.00 (Reference) | 1.00 (Reference) | 1.00 (Reference) |

[a] OR comparing incident ALS cases who ever received a diagnosis code for ALS from a neurologist to those who received a diagnosis code for ALS from only non-neurologists.

[b] OR mutually adjusted for each covariate in the table. Based on ALS cases with complete covariate data (N = 8,451 overall, including 3,605 who received care from a neurologist and 4,846 who did not).

[c] Excludes potential ALS mimics, such as structural spinal pathologies, myasthenia gravis and subtypes, polyneuropathies, and spinal muscular atrophies. OR mutually adjusted for each covariate in the table. Based on ALS cases with complete covariate data (N = 6,781 overall, including 3,605 who received care from a neurologist and 3,176 who did not.)

[d] Other includes Asian, Pacific Islander, Native American, other, and unknown race/ethnicity identification.

[e] Per year.

[f] Based on 2009 residential zip code and 2010 RUCA code. Excludes 13 (0.2%) incident cases with missing RUCA information. A metropolitan area consists of an area with 50,000 or more people; a micropolitan area consists of a population of 10,000 to 49,999; a small town consists of a population of 2,500 to 9,999; and a rural area consists of areas with a population <2,500.

[g] Based on 2009 residential zip code and published list of multidisciplinary ALS clinics in 2013 [28]. Excludes 47 (0.5%) incident cases with missing latitude and longitude to calculate distance.

[h] Based on 2009 residential zip code and ADI. Dichotomized as living in a disadvantaged area if the ADI score was >80 (out of 100). Excludes 122 (1.4%) incident ALS cases with missing ADI information.

Abbreviations: ADI, Area Deprivation Index; ALS, amyotrophic lateral sclerosis; CI, confidence interval; ICD-9-CM, International Classification of Diseases, Ninth Revision, Clinical Modification; OR, odds ratio; RUCA, rural-urban commuting area; SD, standard deviation.

proportion consistent with prior cohorts [31, 32]. Most (91%) beneficiaries lived in a metro-politan/micropolitan area at diagnosis, and 15.5% lived in a disadvantaged area. Our ALS cohort, with and without ALS patients who subsequently received a code for an ALS/MND mimic, had markedly shorter survival when compared to beneficiaries without an ALS code (shown in Fig 1).

### Receipt of neurologist care

Overall, only 3,676 ALS patients (42.9%) ever saw a neurologist (Table 1) during the 5-year fol-low-up. Age was associated with receiving care from a neurologist, with older cases less likely than younger cases to see a neurologist (OR 0.964, 95% CI 0.961–0.968 for each year of age), with effects particularly marked above age 65. Women were less likely than men to see a neu-rologist, but after accounting for all demographic variables, we observed no difference in receiving care from a neurologist according to sex (OR 1.01, 95% CI 0.92–1.10). However, after adjustment for age and other demographics, Black beneficiaries (OR 0.56, 95% CI 0.47–0.67) were less likely to receive care from a neurologist than non-Hispanic White beneficiaries (Table 1). Beneficiaries in a disadvantaged area were less likely to receive care from a neurolo-gist compared to beneficiaries in an advantaged area, even after adjusting for other demo-graphic factors (OR 0.70, 95% CI 0.61–0.80). Beneficiaries who lived ≥50 miles from an ALS center saw a neurologist more often than those closer to an ALS center (Table 1).

### Receipt of practice parameter interventions

Of all beneficiaries with ALS, 26.7% received a feeding tube, 19.2% received NIV, and 15.3% filled a riluzole prescription between diagnosis and death/end of follow-up (Table 2). Patients with ALS who saw a neurologist were much more likely to receive each of these interventions than patients who never saw a neurologist, even after adjusting for patient demographics: feed-ing tube (OR 2.80, 95% CI 2.52–3.11), NIV (OR 10.8, 95% CI 9.28–12.6) and riluzole (OR 7.67, 95% CI 6.13–9.58) (Table 2). Among those who ever saw a neurologist, most of the practice

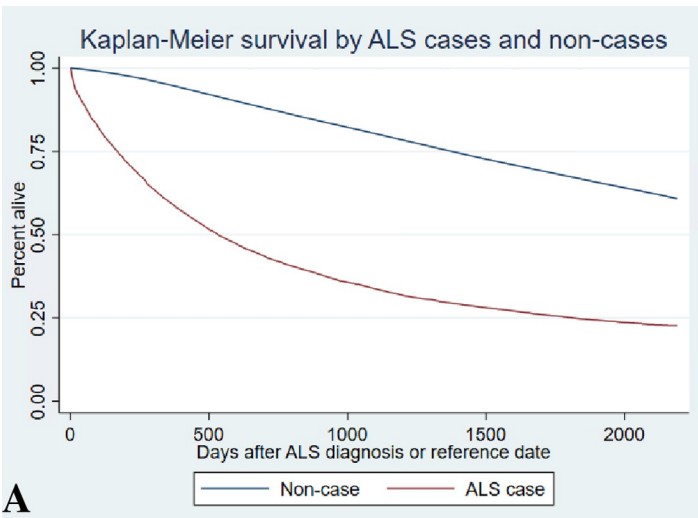
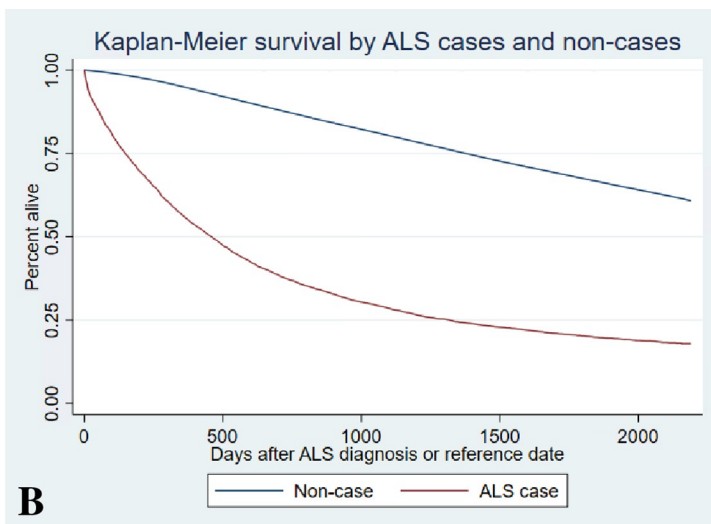

**Fig 1.** Kaplan-Meier Estimates of Survival among Individuals with and without an ALS Code (ICD-9-CM ALS code [335.20]), U.S. Medicare 2009, (A) overall for all incident ALS cases and non-cases in our sample and (B) when excluding potential ALS mimics, such as structural spinal pathologies, myasthenia gravis and subtypes, polyneuropathies, and spinal muscular atrophies. ALS, amyotrophic lateral sclerosis; ICD-9-CM, International Classification of Diseases, Ninth Revision, Clinical Modification.

**Table 2. Use of practice parameter interventions in ALS patients, overall and by receipt of care from a neurologist, U.S. Medicare 2009–2014.**

| All ALS | N = 8,575 n (%) | Received care from a neurologist N = 3,676 n (%) | Only received care from non-neurologists N = 4,899 n (%) | Unadjusted OR (95% CI)[a] | Adjusted OR (95% CI)[a,b] |
|---|---|---|---|---|---|
| Feeding tube[c] | 2,285 (26.7) | 1,421 (38.7) | 864 (17.6) | 2.94 (2.67–3.25) | 2.80 (2.52–3.11) |
| NIV[c] | 1,642 (19.2) | 1,412 (38.4) | 230 (4.7) | 12.7 (10.9–14.7) | 10.8 (9.28–12.6) |
| Riluzole[c,d] | 621 (15.3) | 511 (30.2) | 110 (4.7) | 8.84 (7.11–11.0) | 7.67 (6.13–9.58) |
| **Excluding ALS mimics[e]** | **N = 6,888 n (%)** | **Received care from a neurologist N = 3,676 n (%)** | **Only received care from non-neurologists N = 3,212 n (%)** | **Unadjusted OR (95% CI)[a]** | **Adjusted OR (95% CI)[a,b]** |
| Feeding tube[c] | 2,012 (29.2) | 1,421 (38.7) | 591 (18.4) | 2.80 (2.50–3.12) | 2.66 (2.38–2.99) |
| NIV[c] | 1,556 (22.6) | 1,412 (38.4) | 144 (4.5) | 13.3 (11.1–15.9) | 12.0 (10.0–14.4) |
| Riluzole[c,d] | 585 (19.5) | 511 (30.2) | 74 (5.7) | 7.23 (5.59–9.34) | 6.75 (5.20–8.76) |

[a] OR comparing incident ALS cases who ever received a diagnosis code for ALS from a neurologist to those who received a diagnosis code for ALS from only non-neurologists. Based on all ALS cases with complete covariate data (N = 8,451 overall, including 3,605 who received care from a neurologist and 4,846 who did not; or when excluding ALS mimics, N = 6,781 overall, including the 3,605 who received care from a neurologist and 3,176 who did not).

[b] Adjusted for age (categorical), sex, race/ethnicity (4 categories), urban/rural residency, distance to ALS clinic (categorical), and residence in disadvantaged ADI.

[c] Any time between diagnosis in 2009 and death or end of follow-up at the end of 2014 (5–6 years of follow-up).

[d] Riluzole analysis limited to beneficiaries with available Medicare Part D claims at any point from 2010–2014 and who survived to January 1, 2010 due to availability of claims data (N = 4,050, 47%). Includes 1,692 beneficiaries who ever saw a neurologist (1,653 with complete covariate data) and 2,358 who only ever received care from a non-neurologist overall (2,333 with complete covariate data); or in final column those who received care from a neurologist and 1,310 (1,298 with complete covariate data) who did not.

[e] Excludes potential ALS mimics, such as structural spinal pathologies, myasthenia gravis and subtypes, polyneuropathies, and spinal muscular atrophies.

Abbreviations: ADI, Area Deprivation Index; ALS, amyotrophic lateral sclerosis; CI, confidence interval; NIV, non-invasive ventilation; OR, odds ratio.

parameter interventions occurred after seeing a neurologist: feeding tubes 89.9%, NIV 92.9%, and riluzole 89.8%.

When we compared the other characteristics of ALS patients who did and did not receive the interventions, we observed several potential differences (Tables 3–5). Women tended to receive more care consistent with the practice parameters than men, if they saw a neurologist (all interventions combined, OR 1.40, 95% CI 1.40 1.22–1.61). Black beneficiaries were less likely to receive NIV than non-Hispanic White beneficiaries, whether they saw a neurologist (OR 0.64, 95% CI 0.47–0.88) or not (OR 0.67, 95% CI 0.40–1.15); combined overall (OR 0.52, 95% CI 0.40–0.66). The same was suggested for riluzole regardless of provider type; combined overall (OR 0.86, 95% CI 0.60–1.18) when comparing Black beneficiaries to non-Hispanic White beneficiaries. In contrast, Black beneficiaries were more likely to receive feeding tubes than non-Hispanic White beneficiaries, when they received care exclusively from non-neurologists (OR 1.53, 95% 1.21–1.94). Older beneficiaries received practice parameter interventions markedly less often than younger patients, whether they saw a neurologist or not ($p_{trend} < 0.001$), except for feeding tubes for beneficiaries who only received care from non-neurologists. Urban/rural status in general was not associated with utilization of any of the interventions. Nonetheless, among beneficiaries who saw a neurologist, we observed an inverse association between distance from a multidisciplinary ALS clinic and use of NIV ($p_{trend} = 0.01$). Similarly, we observed that overall beneficiaries in a disadvantaged area received NIV less often than those in an advantaged area, combined (OR 0.75, 95% CI 0.64–0.89). When we accounted for disadvantaged residential area and provider care, beneficiaries who only received care from non-neurologists received feeding tubes more commonly; otherwise, there were no differences in interventions. Finally, there was some indication that regardless of provider, riluzole use was inversely associated with distance to a multidisciplinary clinic, which

**Table 3. ALS patient characteristics and use of feeding tube, by receipt of care from a neurologist, U.S. Medicare 2009–2014.**

| | Received care from a neurologist N = 3,676 | | | Only received care from non-neurologists N = 4,899 | | | Excluding ALS mimics Mutually adjusted OR (95% CI)[a,c] |
|---|---|---|---|---|---|---|---|
| | Feeding tube N = 1,421 % | No feeding tube N = 2,255 % | Mutually adjusted OR (95% CI)[a,b] | Feeding tube N = 864 % | No feeding tube N = 4,035 % | Mutually adjusted OR (95% CI)[a,b] | |
| **Sex** | | | | | | | |
| Male | 50.9 | 59.6 | 1.00 (Reference) | 55.3 | 52.7 | 1.00 (Reference) | 1.00 (Reference) |
| Female | 49.1 | 40.4 | 1.55 (1.35–1.79) | 44.7 | 47.3 | 0.91 (0.78–1.06) | 0.97 (0.81–1.17) |
| **Race/ethnicity** | | | | | | | |
| Non-Hispanic White | 87.5 | 89.1 | 1.00 (Reference) | 80.1 | 86.4 | 1.00 (Reference) | 1.00 (Reference) |
| Black | 6.0 | 5.8 | 0.92 (0.68–1.24) | 12.9 | 8.8 | 1.53 (1.21–1.94) | 1.50 (1.12–2.03) |
| Hispanic | 2.2 | 1.7 | 1.29 (0.77–2.17) | 2.1 | 1.5 | 1.40 (0.81–2.44) | 1.24 (0.63–2.46) |
| Other[d] | 4.4 | 3.4 | 1.34 (0.94–1.91) | 5.0 | 3.3 | 1.76 (1.23–2.52) | 1.89 (1.24–2.88) |
| | | | overall p = 0.12 | | | overall p<0.001 | overall p0.001 |
| **Age, years** | | | | | | | |
| Mean (SD) | 63.2 (12.1) | 65.8 (12.1) | - | 70.1 (12.3) | 70.7 (13.1) | - | - |
| 20–39 | 3.9 | 2.7 | 1.53 (1.01–2.32) | 2.0 | 2.0 | 0.78 (0.44–1.40) | 0.77 (0.39–1.51) |
| 40–49 | 11.1 | 8.4 | 1.42 (1.07–1.89) | 4.2 | 5.4 | 0.63 (0.41–0.97) | 0.60 (0.35–1.02) |
| 50–54 | 8.0 | 6.8 | 1.26 (0.93–1.70) | 5.1 | 4.7 | 0.87 (0.57–1.32) | 0.82 (0.51–1.33) |
| 55–59 | 10.4 | 8.9 | 1.21 (0.92–1.61) | 6.8 | 6.3 | 0.92 (0.63–1.35) | 0.92 (0.59–1.44) |
| 60–64 | 14.9 | 14.9 | 1.00 (Reference) | 10.0 | 8.5 | 1.00 (Reference) | 1.00 (Reference) |
| 65–69 | 19.4 | 16.6 | 1.17 (0.93–1.49) | 16.1 | 15.2 | 0.92 (0.68–1.26) | 0.87 (0.60–1.26) |
| 70–74 | 15.5 | 17.5 | 0.88 (0.69–1.13) | 16.3 | 15.2 | 0.93 (0.69–1.27) | 1.00 (0.69–1.45) |
| 75–79 | 9.6 | 12.8 | 0.74 (0.56–0.97) | 14.8 | 14.9 | 0.89 (0.65–1.21) | 0.93 (0.64–1.36) |
| 80–84 | 5.3 | 7.8 | 0.68 (0.49–0.94) | 15.1 | 13.9 | 0.96 (0.70–1.31) | 0.93 (0.64–1.35) |
| ≥85 | 1.8 | 3.7 | 0.50 (0.31–0.82) | 9.7 | 14.0 | 0.62 (0.44–0.87) | 0.51 (0.34–0.77) |
| | | | p$_{trend}$<0.001 | | | p$_{trend}$ = 0.45 | p$_{trend}$ = 0.38 |
| **Urban-rural residence**[e] | | | | | | | |
| Metro/micropolitan | 89.2 | 91.0 | 1.00 (Reference) | 92.2 | 91.3 | 1.00 (Reference) | 1.00 (Reference) |
| Small town/Rural | 10.8 | 9.0 | 1.15 (0.91–1.46) | 7.8 | 8.7 | 0.87 (0.65–1.16) | 0.88 (0.62–1.26) |
| **Miles from a multidisciplinary ALS clinic**[f] | | | | | | | |
| Mean (SD) | 76 (127) | 74 (146) | - | 81 (209) | 73 (158) | - | - |
| 0–49 | 52.2 | 54.6 | 1.00 (Reference) | 58.8 | 56.8 | 1.00 (Reference) | 1.00 (Reference) |
| 50–99 | 20.8 | 21.5 | 0.99 (0.83–1.19) | 17.3 | 20.3 | 0.82 (0.67–1.01) | 0.83 (0.64–1.07) |
| 100–149 | 14.0 | 12.2 | 1.17 (0.95–1.46) | 10.5 | 12.0 | 0.82 (0.63–1.05) | 0.81 (0.59–1.11) |
| ≥150 | 13.0 | 11.7 | 1.17 (0.93–1.45) | 13.4 | 10.9 | 1.21 (0.96–1.53) | 1.27 (0.95–1.70) |
| | | | p$_{trend}$ = 0.76 | | | p$_{trend}$ = 0.32 | p$_{trend}$ = 0.60 |
| **Residence in a disadvantaged area**[g] | 13.7 | 12.9 | 0.94 (0.76–1.16) | 20.5 | 16.6 | 1.27 (1.04–1.55) | 1.13 (0.88–1.45) |

[a] OR comparing incident ALS cases who ever received a procedure code for a feeding tube, stratified by those who did or did not ever receive an ALS diagnosis code from a neurologist. OR mutually adjusted for each covariate in the table and smoking (not shown, dichotomous).

[b] Based on all ALS cases with complete covariate data (N = 8,451 overall, including 3,605 who received care from a neurologist and 4,846 who did not).

[c] Excludes potential ALS mimics, such as structural spinal pathologies, myasthenia gravis and subtypes, polyneuropathies, and spinal muscular atrophies. Based on those with complete covariate data (N = 6,781 overall, including the 3,605 who received care from a neurologist and 3,176 who did not, including 582 who did not receive care from a neurologist but received a feeding tube).

[d] Other includes Asian, Pacific Islander, Native American, other, and unknown race/ethnicity identification.

[e] Based on 2009 residential zip code and 2010 RUCA code. Excludes 13 (0.2%) incident cases with missing RUCA information. A metropolitan area consists of an area with 50,000 or more people; a micropolitan consists of a population of 10,000 to 49,999; a small town consists of a population of 2,500 to 9,999; and a rural area consists of areas with a population <2,500.

[f] Based on 2009 residential zip code and published list of multidisciplinary ALS clinics in 2013 [28]. Excludes 47 (0.5%) incident cases with missing latitude and longitude to calculate distance.

[g] Based on 2009 residential zip code and ADI. Dichotomized as living in a disadvantaged area if the ADI score was >80 (out of 100). Excludes 122 (1.4%) incident ALS cases with missing ADI information.

Abbreviations: ADI, Area Deprivation Index; ALS, amyotrophic lateral sclerosis; CI, confidence interval; ICD-9-CM, International Classification of Diseases, Ninth Revision, Clinical Modification; NIV, non-invasive ventilation; OR, odds ratio; RUCA, rural-urban commuting area; SD, standard deviation.

**Table 4. ALS patient characteristics and use of an NIV, by receipt of care from a neurologist, U.S. Medicare 2009–2014.**

| | Received care from a neurologist N = 3,676 | | | Only received care from non-neurologists N = 4,899 | | | |
|---|---|---|---|---|---|---|---|
| | NIV N = 1,412 % | No NIV N = 2,264 % | Mutually adjusted OR (95% CI)a | NIV N = 230 % | No NIV N = 4,669 % | Mutually adjusted OR (95% CI)a,b | Excluding ALS mimics Mutually adjusted OR (95% CI)a,c |
| **Sex** | | | | | | | |
| Male | 56.3 | 56.1 | 1.00 (Reference) | 63.0 | 52.7 | 1.00 (Reference) | 1.00 (Reference) |
| Female | 43.7 | 43.9 | 1.06 (0.92–1.21) | 37.0 | 47.3 | 0.75 (0.57–0.99) | 0.81 (0.57–1.15) |
| **Race/ethnicity** | | | | | | | |
| Non-Hispanic White | 89.5 | 87.9 | 1.00 (Reference) | 87.0 | 85.2 | 1.00 (Reference) | 1.00 (Reference) |
| Black | 4.9 | 6.5 | 0.64 (0.47–0.88) | 7.0 | 9.6 | 0.67 (0.40–1.15) | 0.81 (0.42–1.56) |
| Hispanic | 2.0 | 1.9 | 0.95 (0.57–1.60) | 6.1 | 5.1 | 1.09 (0.61–1.96) | 1.27 (0.65–2.50) |
| Otherb | 3.7 | 3.8 | 0.84 (0.58–1.21) | | | | |
| | | | overall p = 0.32 | | | overall p = 0.92 | |
| **Age, years** | | | | | | | |
| Mean (SD) | 62.8 (11.8) | 66.0 (12.2) | - | 65.9 (11.5) | 70.8 (12.9) | - | - |
| 20–39 | 4.0 | 2.7 | 1.46 (0.97–2.20) | 9.6 | 7.1 | 0.65 (0.37–1.13) | 0.65 (0.32–1.32) |
| 40–49 | 11.1 | 8.4 | 1.22 (0.92–1.61) | | | | |
| 50–54 | 8.6 | 6.5 | 1.26 (0.93–1.71) | 7.4 | 4.6 | 0.79 (0.43–1.44) | 0.83 (0.40–1.72) |
| 55–59 | 10.6 | 8.8 | 1.10 (0.84–1.46) | 9.1 | 6.2 | 0.71 (0.40–1.26) | 0.99 (0.51–1.94) |
| 60–64 | 15.7 | 14.4 | 1.00 (Reference) | 16.1 | 8.4 | 1.00 (Reference) | 1.00 (Reference) |
| 65–69 | 19.1 | 16.8 | 1.05 (0.83–1.33) | 15.2 | 15.3 | 0.49 (0.30–0.80) | 0.45 (0.24–0.86) |
| 70–74 | 16.0 | 17.1 | 0.85 (0.67–1.09) | 16.5 | 15.3 | 0.51 (0.32–0.82) | 0.69 (0.39–1.24) |
| 75–79 | 9.8 | 12.7 | 0.72 (0.55–0.94) | 14.8 | 14.9 | 0.47 (0.29–0.76) | 0.48 (0.25–0.91) |
| 80–84 | 5.1 | 12.7 | 0.37 (0.27–0.51) | 11.3 | 28.1 | 0.20 (0.12–0.34) | 0.18 (0.09–0.35) |
| ≥85 | | | | | | | |
| | | | $p_{trend} < 0.001$ | | | $p_{trend} < 0.001$ | $p_{trend} < 0.001$ |
| **Urban-rural residencec** | | | | | | | |
| Metropolitan / Micropolitan | 90.5 | 90.2 | 1.00 (Reference) | 90.0 | 91.6 | 1.00 (Reference) | 1.00 (Reference) |
| Small town / rural | 9.5 | 9.8 | 1.03 (0.81–1.31) | 10.0 | 8.4 | 1.17 (0.74–1.87) | 1.32 (0.75–2.35) |
| **Residence distance (miles) from a multidisciplinary ALS clinicd** | | | | | | | |
| Mean (SD) | 66 (111) | 81 (153) | | 82 (191) | 74 (167) | - | - |
| 0–49 | 56.9 | 51.7 | 1.00 (Reference) | 57.5 | 57.2 | 1.00 (Reference) | 1.00 (Reference) |
| 50–99 | 21.4 | 21.1 | 0.92 (0.77–1.10) | 15.9 | 20.0 | 0.73 (0.49–1.08) | 0.81 (0.50–1.31) |
| 100–149 | 11.9 | 13.5 | 0.79 (0.63–0.98) | 13.3 | 11.7 | 1.01 (0.66–1.54) | 0.91 (0.52–1.61) |
| ≥150 | 9.9 | 13.7 | 0.68 (0.54–0.86) | 13.3 | 11.2 | 1.07 (0.70–1.63) | 1.28 (0.76–2.15) |
| | | | $p_{trend} = 0.01$ | | | $p_{trend} = 0.42$ | $p_{trend} = 0.73$ |
| **Residence in a disadvantaged areae** | 11.9 | 14.0 | 0.87 (0.70–1.08) | 16.4 | 17.3 | 0.89 (0.61–1.30) | 0.91 (0.56–1.48) |

[a] OR comparing incident ALS cases who ever received a procedure code for an NIV, stratified by those who did or did not ever receive an ALS diagnosis code from a neurologist. OR mutually adjusted for each covariate in the table and smoking (not shown, dichotomous).

[b] Based on all ALS cases with complete covariate data (N = 8,451 overall, including 3,605 who received care from a neurologist and 4,846 who did not).

[c] Excludes potential ALS mimics, such as structural spinal pathologies, myasthenia gravis and subtypes, polyneuropathies, and spinal muscular atrophies. Based on those with complete covariate data (N = 6,781 overall, including the 3,605 who received care from a neurologist and 3,176 who did not, including 141 who did not receive care from a neurologist but received an NIV).

[d] Other includes Asian, Pacific Islander, Native American, other, and unknown race/ethnicity identification.

[e] Based on 2009 residential zip code and 2010 RUCA code. Excludes 13 (0.2%) incident cases with missing RUCA information. A metropolitan area consists of an area with 50,000 or more people; a micropolitan consists of a population of 10,000 to 49,999; a small town consists of a population of 2,500 to 9,999; and a rural area consists of areas with a population <2,500.

[f] Based on 2009 residential zip code and published list of multidisciplinary ALS clinics in 2013 [28]. Excludes 47 (0.5%) incident cases with missing latitude and longitude to calculate distance.

[g] Based on 2009 residential zip code and ADI. Dichotomized as living in a disadvantaged area if the ADI score was >80 (out of 100). Excludes 122 (1.4%) incident ALS cases with missing ADI information.

Abbreviations: ADI, Area Deprivation Index; ALS, amyotrophic lateral sclerosis; CI, confidence interval; ICD-9-CM, International Classification of Diseases, Ninth Revision, Clinical Modification; OR, odds ratio; RUCA, rural-urban commuting area; SD, standard deviation.

**Table 5. ALS patient characteristics and use of riluzole[a], by receipt of care from a neurologist, U.S. Medicare 2009–2014.**

| | Received care from a neurologist N = 1,692 | | | Only received care from non-neurologists N = 2,358 | | | Excluding ALS mimics Mutually adjusted OR (95% CI)[a,c] |
|---|---|---|---|---|---|---|---|
| | Riluzole N = 511 % | No Riluzole N = 1,181 % | Mutually adjusted OR (95% CI)[a] | Riluzole N = 110 % | No Riluzole N = 2,248 % | Mutually adjusted OR (95% CI)[a,b] | |
| **Sex** | | | | | | | |
| Male | 49.1 | 53.5 | 1.00 (Reference) | 61.8 | 48.3 | 1.00 (Reference) | 1.00 (Reference) |
| Female | 50.9 | 46.5 | 1.32 (1.06–1.64) | 38.2 | 51.7 | 0.60 (0.40–0.90) | 0.49 (0.30–0.82) |
| **Race/ethnicity** | | | | | | | |
| Non-Hispanic White | 84.0 | 86.4 | 1.00 (Reference) | 85.5 | 82.1 | 1.00 (Reference) | ‒‒[h] |
| Black | 9.4 | 7.0 | 1.22 (0.82–1.81) | 14.6 | 17.9 | 0.72 (0.41–1.26) | |
| Hispanic | 2.7 | 2.7 | 1.01 (0.52–1.96) | | | | |
| Other[b] | 3.9 | 3.9 | 0.97 (0.55–1.69) | | | | |
| | | | overall p = 0.78 | | | overall p = 0.29 | overall p = 0.06 |
| **Age, years** | | | | | | | |
| Mean (SD) | 61.9 (12.1) | 64.3 (12.8) | - | 63.0 (12.7) | 69.3 (13.9) | - | |
| 20–39 | 4.3 | 4.6 | 0.74 (0.41–1.33) | 17.3 | 10.0 | 0.63 (0.33–1.21) | 0.52 (0.25–1.07) |
| 40–49 | 14.5 | 9.8 | 1.26 (0.84–1.91) | | | | |
| 50–54 | 8.2 | 7.1 | 0.96 (0.60–1.54) | 13.6 | 11.5 | 0.44 (0.22–0.88) | |
| 55–59 | 8.6 | 8.6 | 0.82 (0.52–1.30) | | | | |
| 60–64 | 15.7 | 13.0 | 1.00 (Reference) | 21.8 | 8.0 | 1.00 (Reference) | 1.00 (Reference) |
| 65–69 | 19.8 | 19.1 | 0.88 (0.61–1.27) | 29.1 | 31.5 | 0.33 (0.19–0.59) | 0.40 (0.17–0.90) |
| 70–74 | 17.4 | 16.1 | 0.90 (0.62–1.31) | | | | 0.43 (0.19–1.00) |
| 75–79 | 7.4 | 12.2 | 0.48 (0.30–0.75) | 18.2 | 39.0 | 0.17 (0.09–0.31) | 0.19 (0.09–0.41) |
| 80–84 | 4.1 | 9.5 | 0.36 (0.21–0.62) | | | | |
| ≥85 | | | | | | | |
| | | | p$_{trend}$<0.001 | | | p$_{trend}$<0.001 | p$_{trend}$ = 0.001 |
| **Urban-rural residence[c]** | | | | | | | |
| Metropolitan/micropolitan | 90.6 | 89.8 | 1.00 (Reference) | 90.9 | 91.6 | - | ‒‒[h] |
| Small town/rural | 9.4 | 10.2 | 0.95 (0.66–1.38) | - | 8.4 | - | |
| **Residence distance (miles) from a multidisciplinary ALS clinic[d]** | | | | | | | |
| Mean (SD) | 69 (125) | 83 (179) | - | 90 (315) | 67 (131) | - | |
| 0–49 | 58.6 | 52.8 | 1.00 (Reference) | 67.3 | 58.4 | 1.00 (Reference) | 1.00 (Reference) |
| 50–99 | 18.5 | 20.9 | 0.75 (0.56–1.01) | 15.5 | 19.6 | 0.60 (0.34–1.05) | 0.42 (0.24–0.77) |
| 100–149 | 10.6 | 13.5 | 0.69 (0.48–0.98) | 17.3 | 22.0 | 0.62 (0.36–1.08) | |
| ≥150 | 12.4 | 12.8 | 0.90 (0.64–1.27) | | | | |
| | | | p$_{trend}$ = 0.37 | | | p$_{trend}$ = 0.23 | p$_{trend}$ = 0.90 |

(*Continued*)

**Table 5.** (Continued)

| | Received care from a neurologist N = 1,692 | | | Only received care from non-neurologists N = 2,358 | | | |
|---|---|---|---|---|---|---|---|
| | Riluzole N = 511 % | No Riluzole N = 1,181 % | Mutually adjusted OR (95% CI)[a] | Riluzole N = 110 % | No Riluzole N = 2,248 % | Mutually adjusted OR (95% CI)[a,b] | Excluding ALS mimics Mutually adjusted OR (95% CI)[a,c] |
| **Residence in a disadvantaged area[e]** | 15.3 | 14.0 | 1.10 (0.80–1.51) | 17.3 | 19.8 | 0.95 (0.55–1.62) | – –[h] |

[a] Riluzole analysis limited to beneficiaries with available Medicare Part D claims at any point from 2010–2014 and who survived to January 1, 2010 due to availability of claims data (N = 4,050, 47% in our primary riluzole analysis, including 1,692 (46%) total beneficiaries who ever saw a neurologist and 2,358 (48%) who only ever received care from a non-neurologist). OR comparing incident ALS cases who ever received a riluzole medication, stratified by those who did or did not ever receive an ALS diagnosis code from a neurologist. OR mutually adjusted for each covariate in the table and smoking (not shown).

[b] Based on all ALS cases with complete covariate data (N = 3,986 overall, including 1,653 who received care from a neurologist and 2,333 who did not).

[c] Excludes potential ALS mimics, such as structural spinal pathologies, myasthenia gravis and subtypes, polyneuropathies, and spinal muscular atrophies. Based on those with complete covariate data (N = 2,951, overall, including 1,653 who ever saw a neurologist and 1,298 who did not receive care from a neurologist, including 74 who did not receive care from a neurologist but received riluzole).

[d] Other includes Asian, Pacific Islander, Native American, other, and unknown race/ethnicity identification.

[e] Based on 2009 residential zip code and 2010 RUCA code. Excludes 0.1% incident cases with missing RUCA information. A metropolitan area consists of an area with 50,000 or more people; a micropolitan consists of a population of 10,000 to 49,999; a small town consists of a population of 2,500 to 9,999; and a rural area consists of areas with a population <2,500.

[f] Based on 2009 residential zip code and published list of multidisciplinary ALS clinics in 2013 [24]. Excludes 23 (0.6%) incident cases with missing latitude and longitude to calculate distance.

[g] Based on 2009 residential zip code and ADI. Dichotomized as living in a disadvantaged area if the ADI score was >80 (out of 100). Excludes 63 (0.2%) incident ALS cases with missing ADI information.

[h] Not shown due to small numbers.

Abbreviations: ADI, Area Deprivation Index; ALS, amyotrophic lateral sclerosis; CI, confidence interval; ICD-9-CM, International Classification of Diseases, Ninth Revision, Clinical Modification; OR, odds ratio; RUCA, rural-urban commuting area; SD, standard deviation.

strengthened in our sensitivity analyses excluding potential ALS mimics (Table 5). Otherwise, in sensitivity analyses our findings were largely unchanged when beneficiaries with potential ALS mimics were excluded from our cohort (Tables 3–5).

In our mediation analysis, Black race mediated the association between receiving NIV and seeing a neurologist by 43.5% (95% CI 34.1%-63.1%). Seeing a neurologist mediated the association between ADI and NIV by 60.2% (95% CI 39.4%-131%) and riluzole by 100% (95% CI -314%-611%), but the latter was not significant. Seeing a neurologist also mediated the association between age and each intervention: feeding tube by 32.5% (95% CI 23.8%-49.3%), NIV by 37.1% (95% CI 27.9%-49.5%), and riluzole by 52.1% (95% CI 39.9%-76.5%).

## Discussion

In this large, population-based study of all Medicare beneficiaries with incident ALS, we found marked disparities in the use of neurologist care and adherence to AAN practice parameters. Notably, a substantial percentage of patients diagnosed with ALS did not see a neurologist, consistent with Medicare studies in other neurologic diseases [33, 34]. Moreover, Black race, older age, and residence in a socially disadvantaged neighborhood were important determinants of whether patients with ALS received care from a neurologist and independently if they received care consistent with the practice parameters. Further, despite freely available publication of the practice parameters for the care of ALS patients more than five years prior to the end of our study period, neurologists provided care consistent with these evidence-based guidelines far more often than non-neurologists. As interventions are likely more efficacious

in the early stages of the disease [4] and utilization of NIV and feeding tubes can lead to improved survival and quality of life [3], it is critical for providers who take care of ALS patients to be aware of evidence-based ALS practice parameters. Furthermore, newer disease-modifying medicines such as edaravone, toferson, and sodiumphenylbutyrate-taurursodiol (PB-TURSO) are now available, making our findings even more impactful [35, 36].

When we examined differences in utilization for each intervention according to provider type, we found that women underwent feeding tube placement far more often than men, when a neurologist provided care, possibly due to greater prevalence of bulbar onset ALS in women than men [37, 38]. Black beneficiaries received riluzole prescriptions and NIV less frequently than non-Hispanic White patients, despite studies suggesting that Black patients have lower baseline vital capacity and lower baseline functional scores than non-Hispanic White patients at first clinic visit for ALS [12]. Interestingly, Black beneficiaries were more likely to undergo feeding tube placement when compared to non-Hispanic White patients if they did not receive care from a neurologist. We found similar results in beneficiaries who reside in a socially disadvantaged neighborhood when compared to those beneficiaries from more advantaged neighborhoods. We speculate that the differences seen might reflect greater community provider familiarity with ordering feeding tubes, Black beneficiaries' potential slower progressing clinical phenotype [12], and/or lack of early referral/access to neurology for these populations due to SDOH-related health care barriers. Nonclinical risk factors, such as urban versus rural living, are potential sources of diagnostic delay [39, 40], and furthermore, it has been shown that a shortage of neurologists in rural areas delays ALS diagnosis [41]. Although we found no associations related to urban/rural status, we did find less utilization of NIV, and potentially riluzole, with increasing distance to an ALS center, consistent with prior literature [39].

There are likely several reasons that racially and ethnically minoritized beneficiaries, beneficiaries who reside in disadvantaged areas, and elderly individuals receive neurologist care less frequently than other beneficiaries. First, there is a shortage of neurologists in many areas of the U.S., and access may be limited to individuals with high health literacy, transportation, primary care physicians, and access to tertiary care centers. Our findings are congruent with prior studies demonstrating that minoritized beneficiaries tend to be less likely to receive care from specialists when compared to non-Hispanic White beneficiaries [9, 12, 42]. Therefore, the burden of diagnosing and managing these patients with ALS may fall upon non-neurologist providers. Ideally, a patient with a potential ALS diagnosis would be evaluated by a neurologist soon after onset of symptoms, but unfortunately this is not always a possibility, often due to prolonged wait times or social barriers to accessing specialty care [43].

Our study highlights the importance of receiving specialist care for ALS patients and provides evidence that may guide health policies and education efforts in the U.S. Efforts such as ensuring early referral and access to neurologist care could emerge as significant interventions. The focused training on neurologic disease during residency and accumulated clinical experience with ALS likely improved adherence to the practice parameters. Neurologists may also be more successful in the early recognition and management of common ALS-associated comorbidities such as need for feeding tube or NIV. Unfortunately, due to a shortage of neurologists this may not always be possible, thus a more systematic approach to education of non-neurologic providers on basic management and focused on adherence to the practice parameters may improve ALS care. This is of critical importance as the aged population continues to increase, the treatment disparities presented here may continue to worsen without efforts to mitigate them, in particular for Black and disadvantaged beneficiaries.

Despite the study strengths, notably size, representativeness, and comprehensiveness of the claims data used, we acknowledge limitations. It is possible that some of the patients who received a code for ALS, especially those diagnosed by only primary care physicians, were

diagnosed incorrectly. We were unable to verify ALS diagnosis, since the size of this administrative data study and limits of CMS data preclude chart review. In addition, while there are more stringent ALS algorithms for administrative data [13], we chose to include all patients with at least one code for ALS, whether from a neurologist or not, to be able to examine the effect of provider specific utilization. On average it takes one year to receive a diagnosis of ALS from symptoms onset [44], and ALS progresses to death rapidly [45, 46]. Utilizing more than one diagnosis code for ALS would inevitably result in missing individuals who have an ALS code from a non-neurological provider who suspects a diagnosis of ALS but is unable to verify the diagnosis with a neurological specialist due to rapid progression to death or disparities in access to neurological care [47]. In prior work, our group demonstrated that using these stringent criteria would exclude many potential ALS patients and that those beneficiaries with only one ALS code were likely representative of true ALS cases [48]. We would have missed a large number of potential ALS cases if the stringent criteria was applied and limited our analysis. Moreover, we used the simpler, high sensitivity approach in order to ensure that the true ALS (vs. other MND) cases were representative demographically, which is critical for studying health services delivery. We also cannot clearly differentiate subtypes of ALS, like bulbar onset, which might intrinsically lead to differences in interventions used, specifically feeding tubes. Furthermore, in sensitivity analysis excluding potential ALS mimics, despite a reduction of up to 20% of our sample, our key findings were largely unchanged, highlighting the robustness of our findings and cohort.

Notably, difference in adherence to the practice parameters might reflect diagnostic certainty as the disease progresses leading to neurologist referral. However, this would only impact the key findings in our study if these phenotypes differentially presented to neurologists. Furthermore, we were unable to ascertain individual provider or patient preferences for not prescribing or not receiving the practice parameters, leading to potentially perceived decreased practice parameter adherence. This is a particular issue with riluzole, where medication adherence has been reported as low as 64% due to medication side effects or socioeconomic factors such as cost [49]. We also may be missing those beneficiaries prescribed riluzole who had an additional private insurance but not Medicare Part D. We anticipate this is attenuated in part, by restricting our analysis for riluzole to beneficiaries with Medicare Part D. While the practice parameters were published on October 12, 2009 [3] and our cohort is derived from 2009 Medicare data, there might not have been enough time to disseminate the practice parameters amongst providers, which could have a more pronounced effect on non-neurologists. We anticipate this limitation is mitigated, in part, by following beneficiaries for up to five years. Finally, although these findings could be important for health policy in the United States, they might not be generalizable to other systems/countries.

## Conclusion

In summary, we found marked differences in adherence to practice parameters for ALS care depending on provider type and broad indicators of SDOH. This study highlights the importance of neurologist care for ALS patients and the need to overcome barriers and provide care that is more equitable for ALS patients.

## Supporting information

**S1 Checklist. Manuscript strobe checklist–please see attached document: "S1 File. ALS Utilization–STROBE-checklist-v4-combined-PlosMedicine.docx".**
(DOCX)

## Author Contributions

**Conceptualization:** Osvaldo J. Laurido-Soto, Irene M. Faust, Susan Searles Nielsen, Brad A. Racette.

**Data curation:** Irene M. Faust.

**Formal analysis:** Osvaldo J. Laurido-Soto, Irene M. Faust, Susan Searles Nielsen.

**Funding acquisition:** Osvaldo J. Laurido-Soto, Susan Searles Nielsen, Brad A. Racette.

**Methodology:** Osvaldo J. Laurido-Soto, Irene M. Faust, Susan Searles Nielsen, Brad A. Racette.

**Resources:** Brad A. Racette.

**Software:** Irene M. Faust.

**Supervision:** Susan Searles Nielsen, Brad A. Racette.

**Validation:** Irene M. Faust, Susan Searles Nielsen.

**Visualization:** Osvaldo J. Laurido-Soto.

**Writing – original draft:** Osvaldo J. Laurido-Soto, Susan Searles Nielsen.

**Writing – review & editing:** Osvaldo J. Laurido-Soto, Irene M. Faust, Susan Searles Nielsen, Brad A. Racette.

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
