## [Decision Letter · Decision Letter 0]

5 Nov 2023

PONE-D-23-13322Adherence to practice parameters in medical beneficiaries with amyotrophic lateral sclerosisPLOS ONE

Dear Dr. Soto,

Thank you for submitting your manuscript to PLOS ONE. After careful consideration, we feel that it has merit but does not fully meet PLOS ONE’s publication criteria as it currently stands. Therefore, we invite you to submit a revised version of the manuscript that addresses the points raised during the review process.

Thank you for submitting your manuscript to PLOS ONE. After careful consideration, we feel that it has merit but does not fully meet PLOS ONE’s publication criteria as it currently stands. Therefore, we invite you to submit a revised version of the manuscript that addresses the points raised during the review process.

1. Kindly elaborate a little bit on the "right to access to neurological care and treatment to larger groups of US residents", especially minority population groups, the elderly, and other disadvantaged population groups.

2.  Kindly re-evaluate your data to see " if there is a possibility that there are some patients who have ALS diagnosis at first but have other subsequent diagnoses, especially for the patients who didn’t receive the diagnosis from a neurologist." If this is so, consider excluding this category of patients from he analysis, or provide justification and potential impact of their inclusion in the study results.

3.  Kindly re-evaluate the number of patients on "Riluzole" in your own study, and revaluate your results in comparison to other studies in the literature.

4.  Please unify he terms in the Tables within the manuscript, such as “Non-Hispanic White” and “White”, etc.

5. Kindly address all other issues or comments raised by the peer reviewers.

We look forward to receiving your revised manuscript.

Kind regards,

Sylvester Chidi Chima, M.D., L.L.M.

Academic Editor

PLOS ONE

Journal Requirements:

Reviewers' comments:

Reviewer's Responses to Questions

**Comments to the Author**

1. Is the manuscript technically sound, and do the data support the conclusions?

Reviewer #1: Yes

Reviewer #2: Partly

2. Has the statistical analysis been performed appropriately and rigorously? 

Reviewer #1: Yes

Reviewer #2: Yes

3. Have the authors made all data underlying the findings in their manuscript fully available?

Reviewer #1: Yes

Reviewer #2: Yes

4. Is the manuscript presented in an intelligible fashion and written in standard English?

Reviewer #1: Yes

Reviewer #2: Yes

5. Review Comments to the Author

Reviewer #1: This paper has assessed the compliance with the 2009 practice parameters for treatment of ALS patients in the United States, and sociodemographic and provider characteristics associated with adherence. The major findings were that ALS patients treated by neurologists received care consistent with practice parameters more often than those not treated by a neurologist. In addition, Afro-American, older, and disadvantaged beneficiaries received less care consistent with the practice parameters.

Overall, this is a well performed study, demonstrating the inequality of access to health services in US. Its findings are important to promote both a better adherence to practice parameters and a better right to access to neurological care and treatment to larger groups of US residents. Some comments regarding these last points would be useful to increase the impact of the paper.

Reviewer #2: The manuscript titled “Adherence to practice parameters in medical beneficiaries with amyotrophic lateral sclerosis” discussed the influencing factors of patients following the practice parameters. I have the several following concerns which should be addressed by the authors.

1. I wonder if there is a possibility that there are some patients who have ALS diagnosis at first but have other subsequent diagnoses, such as MMN, especially for the patients who didn’t receive the diagnosis from a neurologist. If yes, these patients should be excluded from the analysis.

2. The proportion of patients using riluzole in this study is much lower than previous studies, and even lower than the data reported in developing countries. This result raises doubts about the reliability of the data source, and should be further discussed and analyzed.

3. The terms in the table should be unified, such as “Non-Hispanic White” and “White”.

6. PLOS authors have the option to publish the peer review history of their article (what does this mean?). If published, this will include your full peer review and any attached files.

Reviewer #1: No

Reviewer #2: No

---

## [Author Response · Author response to Decision Letter 0]

6 Dec 2023

We have responded to all comments from the reviewers and editor in the "Response to Reviewers" attached document.

---

## [Decision Letter · Decision Letter 1]

14 Jan 2024

PONE-D-23-13322R1Adherence to practice parameters in medical beneficiaries with amyotrophic lateral sclerosisPLOS ONE

Dear Dr. Soto,

Thank you for submitting your manuscript to PLOS ONE. After careful consideration, we feel that it has merit but does not fully meet PLOS ONE’s publication criteria as it currently stands. Therefore, we invite you to submit a revised version of the manuscript that addresses the points raised during the review process. Please address this direct query from Reviewer 2 as quoted below. And kindly make any corrections on your revised manuscript as necessary: "The author mentioned in manuscript that 19.7% of patients may have ALS-mimics. After excluding these cases, the proportion of patients using Riluzole, NIV and PEG has relatively increased. Is this increase statistically significant? If there is indeed statistical significance, would it be more reasonable to use data that excludes ALS-mimics for analysis?"

We look forward to receiving your revised manuscript.

Kind regards,

Sylvester Chidi Chima, M.D., L.L.M.

Academic Editor

PLOS ONE

Journal Requirements:

Reviewer's Responses to Questions

**Comments to the Author**

1. If the authors have adequately addressed your comments raised in a previous round of review and you feel that this manuscript is now acceptable for publication, you may indicate that here to bypass the “Comments to the Author” section, enter your conflict of interest statement in the “Confidential to Editor” section, and submit your "Accept" recommendation.

Reviewer #1: All comments have been addressed

Reviewer #2: All comments have been addressed

2. Is the manuscript technically sound, and do the data support the conclusions?

Reviewer #1: Yes

Reviewer #2: Yes

3. Has the statistical analysis been performed appropriately and rigorously? 

Reviewer #1: Yes

Reviewer #2: Yes

4. Have the authors made all data underlying the findings in their manuscript fully available?

Reviewer #1: Yes

Reviewer #2: Yes

5. Is the manuscript presented in an intelligible fashion and written in standard English?

Reviewer #1: Yes

Reviewer #2: Yes

6. Review Comments to the Author

Reviewer #1: All my comments have been adequately responded. I plase the authors for their work, which is now fine for publication

Reviewer #2: The author mentioned in manuscript that 19.7% of patients may have ALS-mimics. After excluding these cases, the proportion of patients using Riluzole, NIV and PEG has relatively increased. Is this increase statistically significant? If there is indeed statistical significance, would it be more reasonable to use data that excludes ALS-mimics for analysis?

7. PLOS authors have the option to publish the peer review history of their article (what does this mean?). If published, this will include your full peer review and any attached files.

Reviewer #1: No

Reviewer #2: No

---

## [Author Response · Author response to Decision Letter 1]

23 Apr 2024

We have submitted a "Response to Reviewers 2" document that addresses all comments provided.

We have updated the "Data Availability Statement" to provide more information on the restrictions and how to inquire about obtaining CMS data, including information about the Research Data Assistance center (ResDAC). ResDAC helps manage and obtain CMS data. This is further elaborate din the comments section above and in the Data sharing section in the the "Additional information" tab.

---

## [Decision Letter · Decision Letter 2]

7 May 2024

Adherence to practice parameters in Medicare beneficiaries with amyotrophic lateral sclerosis

PONE-D-23-13322R2

Dear Dr. Soto,

We’re pleased to inform you that your manuscript has been judged scientifically suitable for publication and will be formally accepted for publication once it meets all outstanding technical requirements.

Kind regards,

Sylvester Chidi Chima, M.D., L.L.M, LLD.

Academic Editor

PLOS ONE

Reviewers' comments:

Reviewer's Responses to Questions

**Comments to the Author**

1. If the authors have adequately addressed your comments raised in a previous round of review and you feel that this manuscript is now acceptable for publication, you may indicate that here to bypass the “Comments to the Author” section, enter your conflict of interest statement in the “Confidential to Editor” section, and submit your "Accept" recommendation.

Reviewer #2: All comments have been addressed

2. Is the manuscript technically sound, and do the data support the conclusions?

Reviewer #2: Yes

3. Has the statistical analysis been performed appropriately and rigorously? 

Reviewer #2: Yes

4. Have the authors made all data underlying the findings in their manuscript fully available?

Reviewer #2: Yes

5. Is the manuscript presented in an intelligible fashion and written in standard English?

Reviewer #2: Yes

6. Review Comments to the Author

Reviewer #2: (No Response)

7. PLOS authors have the option to publish the peer review history of their article (what does this mean?). If published, this will include your full peer review and any attached files.

Reviewer #2: No

---

## [Editor Report · Acceptance letter]

23 May 2024

PONE-D-23-13322R2 

PLOS ONE

Dear Dr. Laurido-Soto, 

I'm pleased to inform you that your manuscript has been deemed suitable for publication in PLOS ONE. Congratulations! Your manuscript is now being handed over to our production team.

Kind regards, 

on behalf of

Professor Sylvester Chidi Chima 

Academic Editor

PLOS ONE